# Advancing Multiplex Immunofluorescence Imaging Cell Detection using Semi-Supervised Learning with Pseudo-Labeling

Yasin Shokrollahi[1]                                          YSHOKROLLAHI@MDANDERSON.ORG
Karina Pinao Gonzales[2]                                      KBPINAO@MDANDERSON.ORG
Maria Esther Salvatierra[2]                                   MESALVATIERRA@MDANDERSON.ORG
Simon P. Castillo[1,2]                                        SPCASTILLO@MDANDERSON.ORG
Tanishq Gautam[1]                                            TGAUTAM@MDANDERSON.ORG
Pingjun Chen[1]                                              PCHEN6@MDANDERSON.ORG
B. Leticia Rodriguez[1]                                       BLRODRIGUEZ@MDANDERSON.ORG
Sara Ranjbar[1]                                              SRANJBAR@MDANDERSON.ORG
Patient Mosaic Team[*]                                        SPRABHAKARAN@MDANDERSON.ORG
Luisa M. Solis Soto[2]                                        LMSOLIS@MDANDERSON.ORG
Yinyin Yuan[†1,2]                                            YYUAN6@MDANDERSON.ORG
Xiaoxi Pan[‡1,2]                                             XPAN7@MDANDERSON.ORG

[1] *Institute for Data Science in Oncology, FA1-Quantitative Pathology and Medical Imaging, The University of Texas MD Anderson Cancer Center, Houston, TX.*

[2] *Department of Translational Molecular Pathology, Division of Pathology and Laboratory Medicine, The University of Texas MD Anderson Cancer Center, Houston, TX.*

**Editors:** Accepted for publication at MIDL 2024

## Abstract

Accurate cell detection in multiplex immunofluorescence (mIF) is crucial for quantifying and analyzing the spatial distribution of complex cellular patterns within the tumor microenvironment. Despite its importance, cell detection in mIF is challenging, primarily due to difficulties obtaining comprehensive annotations. To address the challenge of limited and unevenly distributed annotations, we introduced a streamlined semi-supervised approach that effectively leveraged partially pathologist-annotated single-cell data in multiplexed images across different cancer types. We assessed three leading object detection models, Faster R-CNN, YOLOv5s, and YOLOv8s, with partially annotated data, selecting YOLOv8s for optimal performance. This model was subsequently used to generate pseudo labels, which enriched our dataset by adding more detected labels than the original partially annotated data, thus increasing its generalization and the comprehensiveness of cell detection. By fine-tuning the detector on the original dataset and the generated pseudo labels, we tested the refined model on five distinct cancer types using fully annotated data by pathologists. Our model achieved an average precision of 90.42%, recall of 85.09%, and an F1 Score of 84.75%, underscoring our semi-supervised model's robustness and effectiveness. This study contributes to analyzing multiplexed images from different cancer types at cellular resolution by introducing sophisticated object detection methodologies and setting a novel approach to effectively navigate the constraints of limited annotated data with semi-supervised learning.

---

[*] Full list of "Patient Mosaic Team" members is detailed in the Appendix Section D.

[†] Joint corresponding author

[‡] Joint corresponding author

**Keywords:** Semi-supervised Learning, Cell Detection, Computational Pathology, Multiplex Imaging

## 1. Introduction

The introduction of multiplex immunofluorescence (mIF) imaging techniques marks a significant advancement in tissue analysis, enabling the in situ quantification of multiple proteins (Amitay et al., 2023). Cell detection from mIF images is a fundamental task, facilitating the quantification of the tumor microenvironment and spatial configuration with identified cell locations and types, thereby providing insight into tumor progression. Integrating deep learning into medical imaging and pathology has significantly led to advancements in disease diagnosis and treatment strategies. Previous methods mainly formulated cell identification as segmentation under the fully supervised fashion, and cell phenotyping primarily relied on the expression of proteins. Schmidt et al. presented StarDist, a novel method for cell segmentation in microscopy images, utilizing star-convex polygons for segmentation and effectively handling crowded cellular environments (Schmidt et al., 2018). Complementing this, Greenwald et al. developed Mesmer, a deep learning algorithm for whole-cell segmentation in tissue images, which achieved human-level performance and addressed the critical challenge of cell segmentation in tissue imaging (Greenwald et al., 2022). Also, Stringer et al. introduced Cellpose, a generalist algorithm for precise cell segmentation across various microscopy image types without requiring retraining (Stringer et al., 2021). Moreover, groundbreaking advancements have been made by Amitay et al. with the development of CellSighter, a neural network demonstrating over 80% accuracy in cell classification from multiplexed images (Amitay et al., 2023). Additionally, unsupervised methods were also incorporated for cell phenotyping. Bortolomeazzi et al. introduced SIMPLI, a versatile tool for multiplexed image analysis by integrating cell segmentation and classification with unsupervised and supervised methods (Bortolomeazzi et al., 2022).

Despite these advancements, the field still faces challenges concerning the limited availability of comprehensively annotated datasets, heterogeneity across cancer types, and the intricate task of cell segmentation and classification in histology images. Several innovative approaches have been introduced to address the need for improved object detection methods in natural scene images. The study by Abbasi et al. proposed enhancing YOLO's performance on partially labeled datasets by creating pseudo-labels for unlabeled instances, significantly improving generalization performance (Abbasi et al., 2020). Niemeijer et al. developed a method for fusing datasets with partially overlapping classes, employing pseudo-labeling with uncertainty quantification to enhance model robustness (Niemeijer et al., 2023). However, these advanced techniques have not been fully investigated for mIF images.

Tackling the challenges of cell detection posed by tumor heterogeneity and lack of comprehensive annotations in mIF images, our study underscores the critical role of precise quantification of complex cellular patterns in predicting immune cells within the tumor microenvironment—immunotherapeutic responses. We initially trained three detectors on partially annotated datasets and selected the best one based on the performance. Next, we used pseudo labels produced by this model to train a new detector. Furthermore, we comprehensively compared our method's performance across different portions of partial

annotations. To finalize the validation of our pipeline, we performed evaluations on five specific cancer types: Papillary Urothelial Carcinoma (PUC), Penile Squamous Cell Carcinoma (PSCC), Urothelial Carcinoma (UC), Cholangiocarcinoma (CC), and Rectal Squamous Cell Carcinoma (RSCC), all fully annotated by pathologists. It further highlights our approach's robustness, effectiveness, and generalization, establishing a new benchmark in medical image analysis and pathology.

## 2. Methodology

### 2.1. Methodology Overview

To generate fully annotated images, this study delves deeply into medical image analysis, focusing on evaluating the performance of three leading object detection models: Faster R-CNN (Ren et al., 2015), YOLOv5s (Jocher, 2020), and YOLOv8s (Ultralytics, 2023). These models are assessed for their proficiency in detecting and classifying cells within mIF images, a process integral to advancing the precision and speed of object detection, pivotal factors in early and accurate cancer diagnosis. Initially, all detectors were trained using a dataset that was partially annotated. To create this dataset, 10 whole mIF images were used, and patches were extracted from these images for training. Additionally, the dataset was expanded through the use of augmentation techniques. Then, the best performing detector, determined by the comparative analysis, was selected to generate pseudo labels (additional labels generated by the detector with the confidence level greater than 90%) in an iterative training loop. These pseudo labels were then used to retrain a new detector, enhancing the model's learning process and improving its detection capabilities. Figure 1 provides a schematic overview for generating pseudo labels and employing a semi-supervised learning approach for the detection and classification of cells, including immune cells (CD45+), epithelial and cancer cells (panCK+), and others (CD45-panCK-).

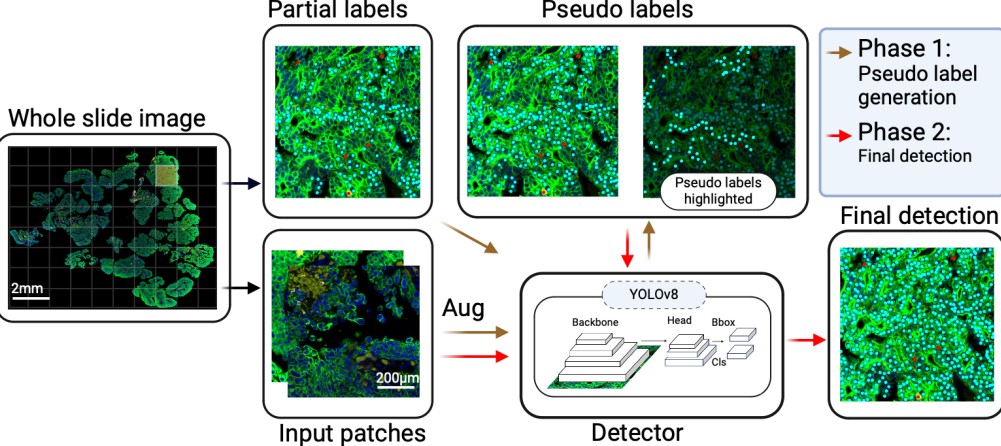

Figure 1: Schematic for generating pseudo-labels and utilizing a semi-supervised learning approach for the detection of cells, including immune cells (CD45+), epithelial and cancer cells (panCK+), and others (CD45-panCK-).

## 2.2. Pre-processing and Annotation Strategy for Training

This study employs an mIF dataset of histology images from papillary urothelial carcinoma tumors of 10 different patients, obtained through our Patient Mosaic initiative following IRB approval, and was generated in line with the GeoMx DSP manufacturer's recommendations (nanoString). Recognizing the labor-intensive nature of annotating large histology images, the study employed a partial annotation strategy. In this study, pathologists annotated approximately 10% of the cells in each of the 10 images, totaling 11,643 cells labeled as CD45, 15,228 cells labeled as panCK, and 9,489 cells labeled as Others. Annotations were made by marking points at the centers of cells for rapid yet accurate identification of each cell type. Subsequently, we drew a fixed square bounding box with a length of 50 pixels around each center point. This step was crucial for converting the point annotations into appropriate inputs for object detection models like YOLO and Faster R-CNN, which necessitate bounding box information. To manage the extensive size of these images, a patch extraction method was applied, creating non-overlapping patches of 640x640 pixels that retain essential cellular features critical for accurate detection (Figure 2). The dataset was further enriched through data augmentation techniques such as flipping, zooming, rotating, and blurring, expanding the dataset to 51,924 patches, of which 70%, 15%, and 15% were used for training, validation, and testing, respectively. The 15% testing portion was carefully chosen from different samples and did not include any augmented images to ensure the model was tested on completely new data.

## 2.3. Model Training, Pseudo Label Generation, and Validation

The pseudo labels generation phase involved pretraining the YOLOv8s model with partially annotated data, guided by semi-supervised object detection strategies to effectively utilize a limited number of labeled images (Gao et al., 2019; Jeong et al., 2019). To further enhance our model's performance and prevent overfitting, we incorporated dropout with a rate of 0.3, extensive data augmentation including mixup and mosaic techniques with ratios of 0.5 and 0.1 respectively, and regularization with an L2 penalty coefficient of 0.01, alongside early stopping with a patience of 5 epochs to ensure generalization during the training process. Moreover, the training was conducted for a shorter span, capped at 50 epochs, to avoid excessive fitting to the partial dataset. This was followed by the generation of pseudo labels, a process inspired by methods such as CSD (Jeong et al., 2019), specifically, we borrowed the idea of using image augmentations, such as flipping, to enforce consistency.

In our approach, pseudo labels were first generated by selecting predictions with a confidence level above 90%. These high-confidence pseudo labels were ensured to contain almost all initial partial annotations from pathologists. This comparison validated the accuracy of all pseudo labels, acting as an additional verification step. Consequently, the method used a confidence threshold for the initial generation and ground truth comparison for validation, ensuring that only the most reliable pseudo labels were added to the dataset.

Although the quality of pseudo labels was not directly evaluated by pathologists, the final model trained with these labels was assessed on the fully annotated dataset. After the pretraining phase with the partial dataset from papillary urothelial carcinoma, we merged the original labels with the pseudo labels (Initially, the annotated dataset by pathologists contained 11,643 cells labeled as CD45, 15,228 cells labeled as panCK, and 9,489 cells

labeled as Others, after adding pseudo-labels generated through semi-supervised learning, the dataset expanded to include approximately 140,000 CD45 cells, 200,000 panCK cells, and 120,000 cells labeled as Others). This enriched dataset was then utilized to retrain the YOLOv8s model. We also analyzed the model's performance at different annotation levels during this retraining phase. Specifically, we considered 100% of the annotations for each class. Subsequently, we experimented with excluding 50% and 75% of the annotations for each class to assess the model's performance under varying levels of annotation scarcity (results presented in the Appendix Section A). The study also included fully annotated datasets of five distinct cancer types, utilized in the final stages of validation to assess the model's performance across varied cancer cell appearances and histology conditions, as illustrated in Figure 2. These datasets were derived from five distinct cancer types, four of which are unseen types during the training process. For each case, four patches of images were fully annotated by two independent pathologists, ensuring high data accuracy. Our primary computational resource was a Linux server equipped with dual NVIDIA GeForce RTX 4090 GPUs, 256GB of random access memory, and a 32-core CPU, ensuring optimal performance and efficiency.

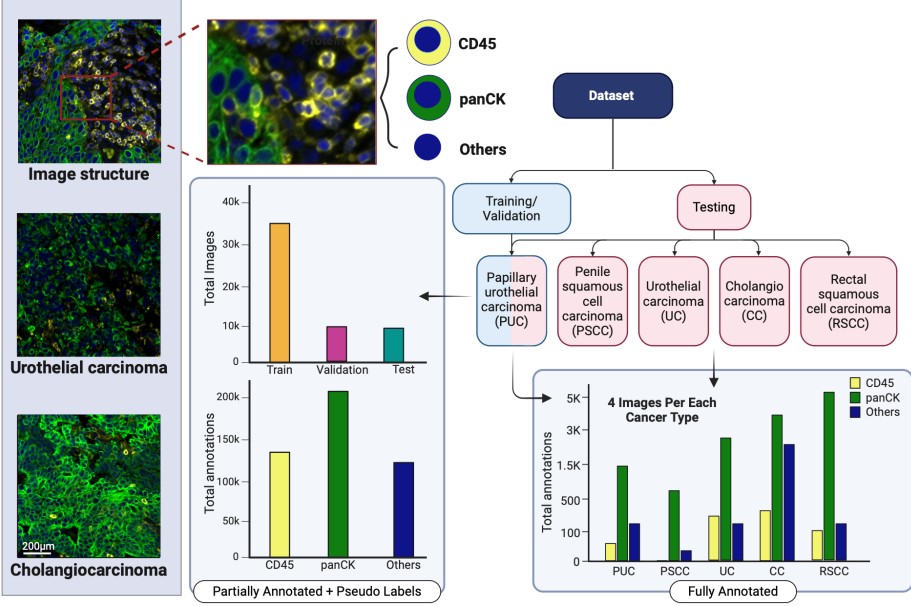

Figure 2: Overview of the dataset showcasing cell morphology, the number of images and annotations per class after augmentation, and fully annotated cancer types for final testing.

## 3. Results

### 3.1. Initial Phase: Selection of Best Object Detection Model

This section encapsulates the comparative performance analysis of various object detection networks, specifically focusing on their capabilities in identifying distinct cell types. After the initial phase, we evaluated the performance of these models, ultimately selecting YOLOv8s for further processes based on its optimal performance. Table 1 provides a

detailed comparison of the object detection models, showcasing YOLOv8s's performance in accurately detecting different types of cancer cells. The table presents key measures of model performance, recall, and mAP50 (mean Average Precision at 50% Intersection Over Union (IoU)), which are central to our evaluation criteria. YOLOv8s achieved excellent recall and mAP50 scores for all types of cells, with recall scores consistently high, ranging from 0.97 to 0.98. Faster R-CNN showed lower performance, with its recall and mAP50 scores varying more widely across different cell types. YOLOv5s also demonstrated high mAP50 and competitive recall scores, nearly matching those of YOLOv8s in certain aspects. However, the comprehensive analysis highlights YOLOv8s's slight edge in performance, especially in terms of recall, marking it as the most accurate and reliable among the tested models.

Table 1: Performance comparison of object detection networks across cell types, trained on an initially partially annotated dataset with 250 epochs.

| Metric | Faster RCNN | | | YOLOv5s | | | YOLOv8s | | |
|--------|------|-------|--------|------|-------|--------|------|-------|--------|
|        | CD45 | panCK | Others | CD45 | panCK | Others | CD45 | panCK | Others |
| Recall | 0.850 | 0.824 | 0.798 | 0.962 | 0.969 | 0.966 | 0.971 | 0.975 | 0.987 |
| mAP50  | 0.846 | 0.815 | 0.791 | 0.983 | 0.985 | 0.984 | 0.985 | 0.993 | 0.988 |

## 3.2. Final Phase: Pseudo Label Integration and Validation on Fully Annotated Dataset

In the final phase, we enhanced YOLOv8s by integrating pseudo labels with the original dataset, which involves semi-supervised learning and iterative refinement. To validate the performance, we used four patches of size 640x640 pixels for each cancer type, which have been fully annotated by two pathologists. The involvement of pseudo labels significantly improved the model's ability to detect cells (Figure 3A, all the $p$-values $< .0001$), validating the effectiveness of the strategy and ensuring the quality of pseudo labels. Figure 3B exemplifies four annotated patches from the PSCC, UC, CC, and RSCC, illustrating the comparison of the cell counts between the pathologists' annotations and the predictions made by YOLOv8s. The comprehensive validation of the YOLOv8s model across various cancer types is presented in Table 2. The model was tested against five cancer types: PUC, PSCC, UC, CC, and RSCC. The model showed high precision, recall and F1 score rates in most categories, with CD45 cells in RSCC and CC demonstrating perfect recall (100%). In the case of PSCC, there were no CD45 cells present in the patches examined, as indicated by the dashes in the table. The model also performed strongly in panCK and Others cell types, especially in PSCC and PUC, showing high precision and recall. Additional details regarding the Area Under the Curve (AUC) metrics ,and the average number of annotations per class for these five types of cancers are provided in the Appendix, Section C, and Section B.

In our study, the annotations provided by pathologists were centered on each cell rather than including segmentation masks. Still, we aspired to compare our results using state-of-the-art cell segmentation methods. Specifically, we applied StarDist (with its "Versatile" model for fluorescent nuclei and the "DSB 2018" model from their 2D paper) (Schmidt et al.,

2018) and Cellpose (using their latest "cyto3" model) (Stringer and Pachitariu, 2024) to predict segmentation masks on our validation datasets. For evaluation, we matched the centers of the predicted segmentations to the bounding boxes of our annotated cell centers. The StarDist method yielded a precision of 0.450, a recall of 0.210, and an F1 score of 0.284, while Cellpose demonstrated a precision of 0.862, a recall of 0.230, and an F1 score of 0.363. Throughout this evaluation, we considered all cells as a single category and utilized only the blue channel of the images.

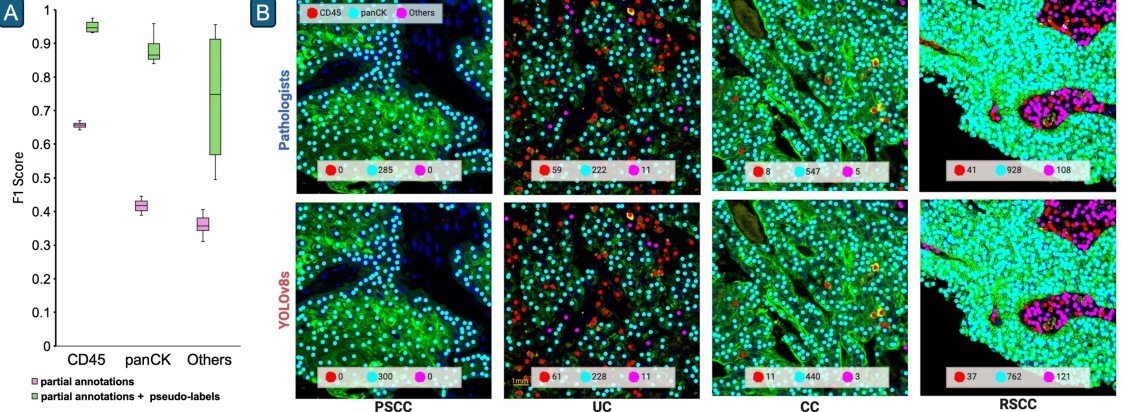

Figure 3: (A): Comparison of YOLOv8s performance on the fully annotated dataset when initially trained on partial annotations and after the integration of pseudo labels, (B): Comparison of cell counts between pathologist annotations and YOLOv8s predictions trained with partial annotations and pseudo labels. The top images show pathologists fully annotated the patches, while the bottom images display YOLOv8's detection. Red dots represent CD45 cells, cyan dots represent panCK cells, and magenta dots indicate Others cell type. The numbers at the bottom denote the count of each cell type.

Table 2: Performance of YOLOv8s on five Cancer Types. The cancer types are indicated as PUC (Papillary Urothelial Carcinoma), PSCC (Penile Squamous Cell Carcinoma), UC (Urothelial Carcinoma), CC (Cholangiocarcinoma), RSCC (Rectal Squamous Cell Carcinoma).

| Type | CD45 | | | panCK | | | Others | | |
|------|-------|-------|-------|-------|-------|-------|-------|-------|-------|
|      | P | R | F1 | P | R | F1 | P | R | F1 |
| PUC  | 1.000 | 0.881 | 0.937 | 1.000 | 0.771 | 0.871 | 1.000 | 0.886 | 0.939 |
| PSCC | - | - | - | 1.000 | 0.891 | 0.943 | 0.386 | 1.000 | 0.557 |
| UC   | 1.000 | 0.921 | 0.959 | 1.000 | 0.734 | 0.847 | 0.452 | 1.000 | 0.623 |
| CC   | 0.880 | 1.000 | 0.936 | 1.000 | 0.751 | 0.858 | 1.000 | 0.719 | 0.837 |
| RSCC | 0.941 | 1.000 | 0.969 | 1.000 | 0.857 | 0.923 | 1.000 | 0.500 | 0.667 |

## 4. Discussion

### 4.1. Interpretation of Results

This study addressed the challenge of partially annotated mIF images through a semi-supervised deep learning approach, focusing on detecting three cell classes: CD45, panCK, and Others. Cell detection in mIF is necessary to quantify and analyze the intricate spatial distribution of cellular patterns within the tumor microenvironment, which is crucial for understanding immunological studies in oncology. The model's validation on fully annotated patches across five cancer types, despite being trained on a dataset with partial annotations (approximately 10% of full annotations), not only underscores its capability to detect the majority of cells within the patches but also demonstrates its versatility in addressing different cancer types. This adaptability and reliability are evident in the consistent performance across various histology conditions, ranging from Papillary Urothelial Carcinoma to Rectal Squamous Cell Carcinoma. The model's adeptness in various contexts reflects findings similar to those of Amitay et al., who underscored the significance of accurate cell classification in computational pathology (Amitay et al., 2023). Furthermore, the ability of YOLOv8s to deliver high performance, even with a limited set of annotations, indicates its robust learning mechanism and capacity to generalize from sparse data. This characteristic is particularly valuable considering the resource-intensive nature of manual annotations in medical imaging. This robustness and adaptability make our model a promising tool in digital pathology, opening new avenues for efficient and accurate cancer cell detection and classification.

### 4.2. Limitations, Future Work, and Broader Implications

While the study presents promising results, certain limitations and opportunities for future research are apparent. This study performed annotations by marking points at the centers of cells rather than using bounding box techniques, it could limit the model's ability to fully understand the spatial context of each cell, which could be critical for certain analytical tasks like cells segmentation. Future endeavors could focus on refining the model's performance under sparse annotations, potentially employing advanced semi-supervised learning techniques, as highlighted by Xu et al. (Xu et al., 2021). Furthermore, extending the model's detection capabilities to encompass a broader range of cell types and pathological conditions could offer a more holistic tool for pathologists, possibly integrating with comprehensive analysis software like SIMPLI, as suggested by Bortolomeazzi et al. (Bortolomeazzi et al., 2022). Also, the absence of CD45 marker values in PSCC samples due to the non-availability of CD45 in PSCC patches points to a limitation in evaluating the model's capability to detect the CD45 marker in PSCC, reflecting a need for more comprehensive data to validate the model's performance for this specific marker. Moreover, while our primary objective was to identify cell centers and cells detection, comparisons with state-of-the-art methods such as StarDist and Cellpose highlight the potential for future research to enhance cell annotations by incorporating segmentation masks, thereby potentially improving the performance metrics against state-of-the-art segmentation methods.

In summary, our study contributes to mIF imaging analysis by introducing a semi-supervised learning model specifically designed for cell detection across different cancer

types. By integrating a novel pseudo-label generation strategy, we aim to improve detection accuracy. This method showcases the potential of the YOLOv8s model to address the unique challenges presented by mIF datasets, potentially moving us toward enhanced precision and efficiency in healthcare diagnostics.

## Data and Code Availability

The sample dataset, source code, and any additional materials supporting the findings of this study are available in the following GitHub repository: https://github.com/idso-fa1-pathology/semi-supervised-cell-detection

## Acknowledgments

We thank members of the Yuan lab for the critical discussion. We thank Dr's M. Neus Bota and Sabitha Prabhakaran for their assistance with Patient Mosaic. This work was supported by the MD Anderson Patient Mosaic™ Project at The University of Texas MD Anderson Cancer Center. Patient Mosaic is supported by generous philanthropic contributions from the Albert and Margaret Alkek Foundation, among others. This project is funded by Lyda Hill Philanthropies.

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

## Appendix A. Annotation Impact: Analyzing Model Performance at Different Annotation Levels

Given the superior performance of YOLOv8s in detecting cells, as evidenced in Table 3, we further investigated its robustness under varying annotation scenarios. Specifically, we evaluated the model's performance when trained with different proportions of annotated data: 100%, 50%, and 25%. The results of this experiment, detailed in Table 2, provide insight into the minimum level of annotations required to maintain acceptable detection results. As presented in Table 3, YOLOv8s maintain respectable performance metrics with a full annotation set (100%). However, a notable decrease in precision, recall, and mAP50 is observed as the level of annotations is reduced. With 50% annotations, the model achieves a precision of 0.721 and a recall of 0.743, alongside a mAP50 of 0.797, indicating a moderate decline in performance. The impact is more noticeable at 25% annotations, where precision drops to 0.193, recall to 0.515, and mAP50 to 0.182, suggesting a significant compromise in the model's detection capabilities. It is important to note that our strategy for reducing annotations was methodical. We ensured an even distribution of annotations across different cell types rather than removing them randomly. This approach was intended to maintain a balanced representation of each cell type in the training data. These findings highlight the

importance of enough annotations for training robust deep-learning models like YOLOv8s. While the model demonstrates a certain degree of tolerance to reduced annotations, ensuring a higher percentage of annotated data is imperative for optimal performance, especially in the critical domain of cancer cell detection.

Table 3: YOLOv8s performance with varying annotation levels (250 epochs.)

| Metric | 100% Annotations | 50% Annotations | 25% Annotations |
|---|---|---|---|
| Precision | 0.986 | 0.721 | 0.193 |
| Recall | 0.967 | 0.743 | 0.515 |
| mAP50 | 0.989 | 0.797 | 0.182 |

## Appendix B.  Average Annotation Comparison Between Pathologists and YOLOv8s

Figure 4 illustrates the average number of annotations divided into CD45, panCK, and Others identified by pathologists and the YOLOv8s algorithm in five cancer types. The chart shows that pathologists annotate panCK markers more frequently than YOLOv8s in all cancer types, with a particularly high average in RSCC and PSCC. In contrast, for CD45, pathologists and YOLOv8s have a lower and closely matched average number of annotations. The Others cells show a variable pattern, with pathologists generally identifying more annotations except in CC, where YOLOv8s has a higher count. This visualization represents the comparison between pathologists' manual annotations and automated annotations by the YOLOv8s algorithm, emphasizing the differences and potential areas for algorithmic improvement or training.

## Appendix C.  Model Performance Evaluation via AUC Metrics

We assessed our model's classification capabilities for three cell types—CD45, panCK, and Others—using Receiver Operating Characteristic (ROC) curves and AUC metrics, against pathologist-provided ground truth annotations. This approach highlights our final model's precision in identifying each cell type within cancerous tissues, illustrating the balance between sensitivity and specificity.

The model demonstrated variable accuracy across cell types, indicated by the AUC scores: CD45 achieved a high score of 0.90, showing excellent model performance in identifying CD45 cells. In comparison, panCK and Others cell types scored 0.80 and 0.83, respectively, suggesting areas for improvement.These findings underscore the potential of our model in supporting pathologists by rapidly and accurately classifying cellular structures in cancerous tissues.

## Appendix D.  Patient Mosaic Team Members

For the review process, the names of individuals in the Patient Mosaic Team have been omitted.

The following individuals are members of the Patient Mosaic Team: Nadim J Ajami, Azad Ali, Franklin Alvarez, Brittany Alverez, Bianca Amador, Surosh Avandsalehi, Claudia

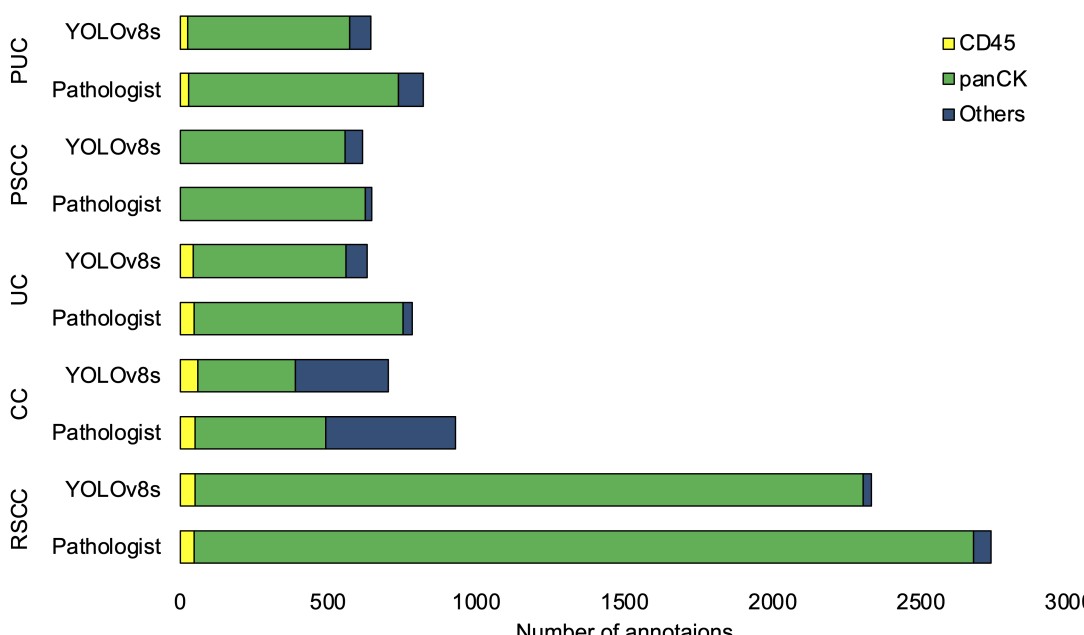

Figure 4: Average annotation comparison between pathologists and YOLOv8s for CD45, panCK, and Others in 5 distinct cancers type.

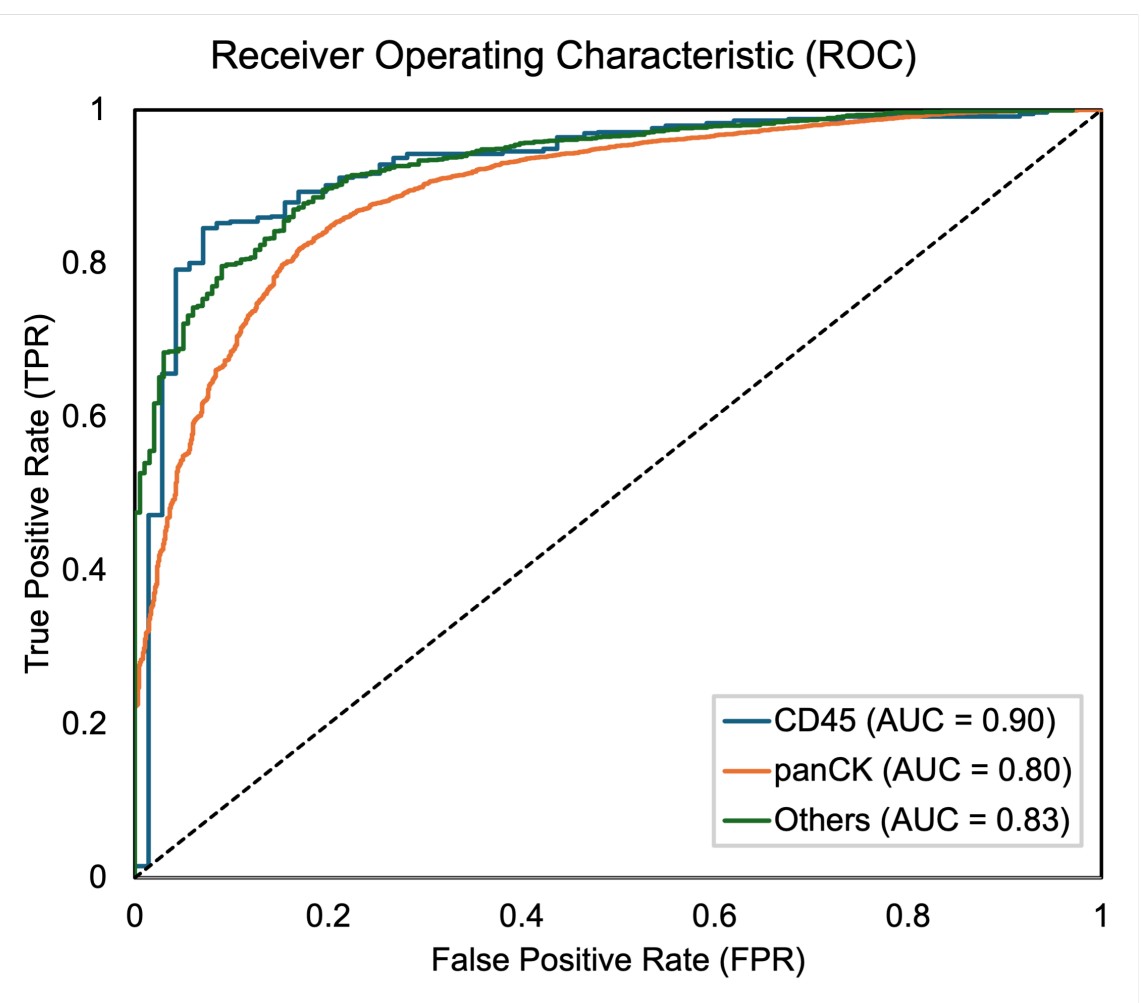

Figure 5: ROC curves and AUC scores for CD45 (0.90), panCK (0.80), and Others (0.83), reflecting the model's classification efficacy for three cell types. Each curve delineates the trade-off between the True Positive Rate and the False Positive Rate, offering a quantifiable accuracy measure.

Alvarez Bedoya, Katrice Bogan, Elena Bogantenkova, Elizabeth Bonojo, Maria Neus Bota-Rabassedas, Elizabeth M Burton, Noble Cadle, Vanessa Castro, Chi-Wan Chow, Randy Aaron Chu, Candace Cunningham, Carrie Daniel-MacDougall, Nana Kouangoua Diane C, Mary Domask, Sheila Duncan, Andrew Futreal, Vivian Gabisi, Jessica Gallegos, Andrea Galvan, Ana Garcia, Jose Garcia, Celia Garcia-Prieto, Christopher Gibbons, Jonathan Benjamin Gill, Dominic Guajardo, Curtis Gumbs, Kristin J Hargraves, Tim Heffernan, Joshua Hein, Sharia Hernandez, Charlotte Hillegass, Yasmine M Hoballah, Theresa Honey, Chacha Horombe, Habibul Islam, Stacy Jackson, Jeena Jacob, Akshaya Jadhav, Robert Jenq, Weiguo Jian, Juliet Joy, Isha Khanduri, Walter Kinyua, Laura Klein, Mark Knafl, Larisa Kostousov, Ying-Wei Kuo, Wenhua Lang, Barrett Craig Lawson, Alexander Lazar, Jack Lee, Erma Levy, XiQi 'Cece' Li, Latasha D Little, Yang Liu, Yan Long, Vielka Lopez, Wei Lu, Sandra Lugo, Aaliyah Maldonado, Jared Malke, Asri Margono, Dipen Maheshbhai Maru, Grace Mathew, Brian McKinley, Jennifer Leigh McQuade, Courtney McRuffin, Gertrude Mendoza, Christopher Miller, Raymond Montoya, Francisco Motemayor, Theresa Nguyen, Heather Perez, Juan Posadas Ruiz, Sabitha Prabhakaran, Mallory Psenda, Gabriela Raso, Mike Roth, Pranoti Sahasrobhojane, Amber Savant, Keri L Schadler, Alejandra Serrano, Kenna R Shaw, Julie M Simon, Elizabeth Sirmans, Luisa Maren Solis Soto, Xingzhi 'Henry' Song, Meghan Stennis, Huandong 'Howard' Sun, Maria Chang Swartz, Marialeska Tariba-Edick, Christopher Vellano, Angela Walker, Ignacio Ivan Wistuba, Scott Eric Woodman, DeArtura Young, Jianhua 'John' Zhang, Haifeng Zhu, Hui 'Helen' Zhu, Olga Bat, Shadarra Crosby, Ellie Freebern, Cindy Hwang, Diana Kouangoua, Yang Li, Sharon Miller, Xiaogang 'Sean' Wu.

