# OpenReview forum: "Advancing Multiplex Immunofluorescence Imaging Cell Detection using Semi-Supervised Learning with Pseudo-Labeling"
_MIDL.io/2024/Conference — MIDL 2024 Poster_

### Official Review · Reviewer_YHMw · 2024-02-26

**Confidence:** 4
**Preliminary Rating:** 1
**Recommendation:** Poster
**Final Rating:** 2.5

**Summary:**

In this paper, the authors propose to enhance cell detection in low-data regimes for multiplex immunofluorescence imaging by using self-distillation. They train three different detection models on a small manually annotated dataset and chose YOLOv8s of these as the best-performing one. In the next step, they predict unseen data by the model to generate pseudo-labels. Another YOLOv8s model is trained using the combination of manual and pseudo-labels. The authors evaluate the resulting model on fully, manually annotated test data of 5 different cancer types.

**Strengths:**

* Utilizing little manually annotated training data is a relevant topic to all medical imaging problems
* The authors highlight that the test data is taken from different samples than training data
* Testing is based on 5 different cancer types of which 4 have not been included in training

**Weaknesses:**

* the paper is not very well structured. Examples are
  * in the introduction, related work for cell segmentation is discussed (Greenwald et al), then cell classification (Amitay et al. etc) and then segmentation again (Schmidt et al. etc).
  * The results section contains a lot of description on the evaluation process, which would belong to an evaluation section and a lot of result interpretation, which would belong to the discussion section
* There is quite some redundancy in the paper
* The experiment of the different annotation levels is entirely in the appendix, still its results are referred to in the discussion
* In the whole paper it is not clear whether the model performed only detection of cells or also classification. Are there 3 different classes the model has to decide between or is just the evaluation separated into those classes?
* The results of the proposed approach are not compared to any other approaches. Comparison to state-of-the-art methods or comparison to the YOLOv8s model without training on the pseudo-labels would have been useful
* The paper lacks novelty, as the methods are already state of the art (as the authors explain in the introduction). Also, the validation is very limited such that there are only limited new findings on the application of the method on mIF images

**Detailed Comments:**

* The references to Hinton et al, is at the wrong position, it should be after the term "Self-distillation" and not after its application to immune infiltration
* Evaluation metrics are not explained. mAP50 should at least have one sentence of explanation or a reference to one
* The manual annotations should be described in some detail. Is it point or bounding box annotations? How do you annotate 10% of the data? Is it 10% of the images or 10% of the cells? If 10% of the cells, how can false positives be measured in order to calculate precision?
* "Furthermore, the study incorporated additional fully annotated datasets of five cancer types for final validation, enriching the robustness and adaptability of the models to different cancer cell appearances and histology conditions". This is very imprecise language. Performing some final validation by definition does not change the model and therefore cannot enrich robustness and adaptability of the models.
* "We evaluated the generated pseudo labels through a designed loop by comparing them with the ground truth." is this just the description of how evaluation is performed? Otherwise, it is not clear what exactly is compared and what the result and the consequences of the comparison is
* The authors state that YOLOv5s performs similar to YOLOv8s, while in the very next sentence they claim that the results highlight YOLOv8s' superiority. This is contradicting
* The reference to Qu et al in the Conclusion is redundant and already done in the introduction

**Justification Of Final Rating:**

My 2 major concerns were addressed in the last revision. Regarding the contributions: Thank you for pointing out the differences between your apporach and Abbasi et al. This is now more clear to me. Regarding the comparison to SOTA: Thank you for the comparison with CellPose and StarDist. While the the comparison with these methods is useful (although being segmentation instead of detection, but requested by another reviewer) it is not a fair comparison as the Yolo model was trained with the specific training dataset while StarDist and CellPose were pretrained on different data. Now that your contribution is an advancement over Abbasi et al., it would be necessary to also compare with this method. Otherwise we cannot tell whether your approach actually is an improvement.

**Justification Of The Preliminary Rating:**

The contributions of the paper does not become clear. It does not present a novel method nor does it validate an existing method thoroughly. Additionally, the structure of the writing is often confusing, redundant, and lacking important details.

**Questions To Address In The Rebuttal:**

The authors should be very clear on the contributions of their work and support their claims by performing the required evaluations. A rewrite of their paper following a clear structure would then help to get this message across

**Special Issue:**

No

---

> ### Author Response · Authors · 2024-03-17
> **Clarification and Enhancement of Methodology for Cell Detection in mIF Imaging**
>
> We are thankful to Reviewer 3 for their detailed review and constructive feedback. Below, we address the raised concerns with modifications and clarifications implemented in our revised manuscript:
>
> 1. Manuscript Structure and Clarity:
> Thank you for your valuable feedback.
> - We have carefully restructured the introduction section to ensure that all related literature is presented cohesively and consistently.
> - The results section has been restructured to clearly separate the evaluation process and interpretation of results, aligning content more appropriately with the sections.
> - Redundancies identified across the paper have been meticulously removed to streamline content and improve readability.
> - Details on the experiment of different annotation levels, previously in the appendix, have been refined to ensure a more integrated discussion within the main text.
> 2. Model's Functionality (Detection and Classification):
> Thanks for your comments, our model is designed for both detection and classification of cells. As illustrated in Figure 1, the model differentiates among three categories: immune cells (CD45+), epithelial and cancer cells (panCK+), and others (CD45-panCK-). This clarifies that the evaluation encompasses these specific classes, detailing our method's capacity to not only detect but also accurately classify cells into these predetermined categories.
> 3. Comparison to Other Approaches and with partial annotaions:
> - Thank you for your feedback. As suggested by the reviewer, to evaluate the contribution of pseudo labels to improve the task performance across the five cancer types, in this version we compare the performance of models with and without pseudo labels. We show that incorporating pseudo labels generation significantly improves F1 score (CD45: 0.656 vs. 0.949, panCK: 0.417 vs. 0.880, Others: 0.359 vs.0.740) with p values (Wilcoxon test) of 3.0517578125e-05, 1.9073486328125e-06, 3.0517578125e-05, respectively. We have incorporated these results in Figure 3A and section 3.2, Also, we have added the Area Under the Curve (AUC) performance metrics into Appendix C and referenced it in Section 3.2 of the results for a more thorough evaluation of the model's discriminative capacity.
> 4. The references to Hinton et al, is at the wrong position, it should be after the term "Self-distillation" and not after its application to immune infiltration.
> Thanks, the reference location has been modified
> 5. Evaluation metrics are not explained. mAP50 should at least have one sentence of explanation or a reference to one.
>  Thank you. We have added this part (mean Average Precision at 50% Intersection Over Union (IoU)) to the sentence: “The table presents key measures of model performance, recall, and mAP50 (mean Average Precision at 50% Intersection Over Union (IoU)), which are central to our evaluation criteria.”
> 6. The manual annotations:
> Thank you for your valuable feedback. To clarify the annotation and model training process further, we have updated the methods section of our manuscript as follows: After pathologists annotated the center of cells by marking approximately 10% of the cells across 10 images, we proceeded to draw 50*50 pixels bounding box around each point. In total, pathologists annotated 11,643 cells as CD45+, 15,228 as panCK+, and 9,489 as Others. This step was crucial for adapting the point annotations for training object detection models, which require bounding box inputs. For the evaluation of our model, we compared the predicted bounding boxes with this ground truth bounding boxes to accurately measure false positives rate and calculate precision, ensuring the model's effectiveness in detecting and classifying cell types.
> 7. Contributions and Structure:
> - We have revised the sentence for clarity: "The study also incorporated fully annotated datasets from five distinct cancer types during the final validation phase. This was to evaluate the model's ability to perform across a range of cancer cell appearances and histological conditions."
> -  Thanks for great feedback, we have updated Section 2.3 to address concerns about parameter tuning, pseudo-label quality, and overfitting. Our approach ensures reliable pseudo labels by including only high-confidence predictions (over 90%). The final model's performance across five distinct cancer types, which were fully annotated by expert pathologists not seen during training, attests to our method's robustness and effectiveness against overfitting.
> - Thanks for feedback, kindly we revised to this sentence to avoid contradiction: “YOLOv5s also demonstrated high mAP50 and competitive recall scores, nearly matching those of YOLOv8s in certain aspects. However, the comprehensive analysis highlights YOLOv8s's slight edge in performance, especially in terms of recall, marking it as the most accurate and reliable among the tested models.”
> - Thanks for the feedback; we removed the repetitive reference.

---

> > ### Comment · Reviewer_YHMw · 2024-03-27
> >
> > Thank you for your revision. Several of my concerns have been addressed. However, there are still two major issues: Firstly, the paper lacks a comparison with other state of the art. Secondly, the contributions are still not clear. They are stated in 2 places in the paper and read like if the authors have invented ("[...] introducing an advanced semi-supervised learning mode [...]") this semi-supervised approach while at the same time crediting Abbasi et al. for this. In my view, the authors have only applied this existing technique to a new dataset. However, for a validation paper, more evaluation would have been required (such as comparison to SOTA)

---

> > > ### Author Response · Authors · 2024-03-27
> > > **The response details improvements to the manuscript, including comparative analysis with SOTA methods and clarifications of unique contributions, highlighting methodological in pseudo-label generation and domain-specific training for mIF data analysis.**
> > >
> > > We are thankful for your great detailed review and constructive feedback, we address the raised concerns with modifications and clarifications implemented in our new revised manuscript regarding to SOTA and clarification of our contributions:
> > >
> > > 1. Lack of Comparative Analysis with SOTA Methods:Thanks for the great feedback and suggestions. We have improved our manuscript with additional experiments and discussions to address the concerns raised. We have included results from StarDist and Cellpose in Section 3.2 of our manuscript. These experiments were conducted applying their pre-trained models to our validation dataset. Specifically, we utilized only the blue channel for prediction and treated all cells as a single class to align with our research context. The performance metrics for these methods were evaluated by aligning the centers of predicted segmentation masks within the bounding boxes of annotated cell centers, acknowledging the absence of segmentation masks in our ground truth.
> > > The results from StarDist and Cellpose are now detailed in the manuscript, indicating a precision of 0.450, a recall of 0.210, and an F1 score of 0.284 for StarDist, whereas Cellpose achieved a precision of 0.862, a recall of 0.230, and an F1 score of 0.363. These findings confirm our decision to focus on detection methods, like YOLO, optimized through our training pipeline with only partial annotations. Furthermore, they underscore the challenges and concerns when applying segmentation methods to our specific dataset.
> > > In the discussion and limitations sections, we have expanded upon the implications of these comparisons. We acknowledge our primary aim of identifying cell centers and detection and juxtaposing our approach with state-of-the-art methods,  StarDist and Cellpose, validateing our methodology's efficacy and opening routes for future research. This includes integrating segmentation masks to enhance cell annotations and improving comparative performance against leading segmentation methods.
> > > | Method   | P     | R    | F1    |
> > > |----------|-------|------|-------|
> > > | Stardist | 0.450 | 0.210| 0.284 |
> > > | Cellpose | 0.862 | 0.230| 0.363 |
> > > | YOLOv8s | 0.904 | 0.850| 0.847 |
> > >
> > > 2. clarification of contribution:
> > > Thanks for great feedback, we appreciate the opportunity to clarify the technical distinctions between our approach and the work of Abbasi et al. on enhancing object detection models with semi-supervised learning and pseudo-label generation. Our methodology, while inspired by the broad principles outlined by Abbasi et al., incorporates several key innovations and domain-specific adaptations that distinguish it significantly:
> > > - Pseudo-Label Generation Strategy: Abbasi et al. employed a framework that relied on object detectors pre-trained (proxy network) on individual datasets to generate a set of pseudo-labels for unlabeled instances. Our method differentiates with this by using an iterative pseudo-labeling strategy tailored for mIF data. This involves dynamically generating pseudo-labels based on model confidence (> %90 confidence rate) and validation on fully annotated dataset annotated by pathologists.
> > >
> > > - Training Process: Our training process incorporates domain-specific augmentations and optimizations to improve detection performance across diverse cancer types, which is critical for applications in precision medicine and pathology research.
> > >
> > > - Also based on your suggestion we changed that sentence “this research contributes to the field of mIF image analysis by introducing an advanced semi-supervised learning model for cell detection, paving the way for future explorations and practical implementations” to “our study contributes to mIF imaging analysis by introducing a semi-supervised learning model specifically designed for cell detection across different cancer types. By integrating a pseudo-label generation strategy, we aim to improve detection accuracy.”

---

### Official Review · Reviewer_aDgm · 2024-02-28

**Confidence:** 5
**Preliminary Rating:** 3
**Recommendation:** Poster
**Final Rating:** 4

**Summary:**

This paper discusses a semi-supervised approach developed to improve cell detection in mIF within the tumor microenvironment. The challenge arises from limited and unevenly distributed annotations for training cell detectors. The authors tested three object detection models with tremendous partially annotated data from different cancer types. An enriched dataset was created using pseudo labels generated by YOLOv8s. The fine-tuned model achieved high accuracy on fully annotated data from five cancer types, demonstrating the effectiveness of the semi-supervised approach in cellular analysis of mIF.

**Strengths:**

* The annotation is incredibly large.
* The paper covers various diseases with clinical relevance.
* The model selection is easy to follow and should aid in the implementation of other potential marker combinations.

**Weaknesses:**

* The representation could benefit from some improvement. For instance, the description accompanying Figure 1 may require enhanced clarity to better convey the process of training all detectors on partially annotated datasets and the subsequent selection of the separate top-performing model for pseudo label generation through an iterative loop.

* The paper repeatedly claims that the improvement is significant. However, there is no statistical significance test provided as evidence.

**Detailed Comments:**

* Inclusion of the Area Under the Curve (AUC) performance metrics in the appendix is recommended for a more comprehensive evaluation of the model's discriminative ability.

* Concerning the generation of pseudo labels and the potential for cascading errors, it would be prudent for the authors to outline the measures implemented to mitigate error propagation from the pseudo label generator to the final model. Rigorous cross-validation, confidence thresholding, or incorporating expert review in iterations could be potential strategies to ensure the reliability and accuracy of the data used for further training.

* Figure 1 would benefit from adding the unit of measurement, specifically micrometers, to clarify the scale of observation.

**Justification Of Final Rating:**

My initial comments were addressed satisfactorily; however, the experimental approach to testing the mitigation strategies for pseudo label inaccuracies remains partially unexplored. To more rigorously evaluate these strategies, I recommend comparing the pseudo labels with pathologist-generated labels in a small, targeted region. This comparison could provide deeper insight into the effectiveness of the pseudo labeling process. Although this suggestion might extend beyond the current scope of your study, it is crucial to understand how discrepancies in pseudo labeling could potentially compromise the entire training process. Detailed exploration and documentation of such scenarios would significantly strengthen the manuscript.

**Justification Of The Preliminary Rating:**

The need for improved clarity in figures and methodological rigor, including statistical validation of significant claims.

The importance of addressing the propagation of errors from pseudo labels to the final model.

I also suggest providing additional quantitative metrics, such as AUC performance, to substantiate the paper's findings.

**Questions To Address In The Rebuttal:**

Please review the weaknesses and detailed comments sections. Cross-validation is encouraged but not mandatory during the rebuttal phase.

**Special Issue:**

No

---

> ### Author Response · Authors · 2024-03-17
> **Addressing Feedback for Improved Cell Detection in mIF Imaging**
>
> We are grateful for Reviewer 2’s comprehensive review and constructive suggestions. Our revised manuscript has been updated to address the points raised, as outlined below.
>
> 1. Clarity and Representation Improvements:
> In response to the feedback on enhancing the clarity of Figure 1, we have revised the description to more effectively convey the process of training detectors on partially annotated datasets and the selection process for the top-performing model used in pseudo label generation. This clarification aims to improve the reader's understanding of our methodology and its iterative nature.
> 2. Statistical Significance of Improvements:
> Thank you for your feedback. As suggested by the reviewer, to evaluate the contribution of pseudo labels to improve the task performance across the five cancer types, in this version we compare the performance of models with and without pseudo labels. We show that incorporating pseudo labels generation significantly improves F1 score (CD45: 0.656 vs. 0.949, panCK: 0.417 vs. 0.880, Others: 0.359 vs.0.740) with p values (Wilcoxon test) of 3.0517578125e-05, 1.9073486328125e-06, 3.0517578125e-05, respectively. We have incorporated these results in Figure 3A and section 3.2, Also, we have added the Area Under the Curve (AUC) performance metrics into Appendix C and referenced it in Section 3.2 of the results for a more thorough evaluation of the model's discriminative capacity.
> 3. Inclusion of AUC Performance Metrics:
> Thank you for your valuable feedback. As suggested, in this version we have incorporated the AUC metrics (0.90, 0.80, 0.83 for CD45, panCK, and Others, respectively) into Appendix C and referenced it in Section 3.2 of the results for a more thorough evaluation of the model's discriminative capacity.
> 4.Mitigating Error Propagation from Pseudo Labels:
> Thank you for your great feedback. In this version we have clarified the details related to parameter tuning, pseudo-label quality, and overfitting. We ensured reliability of pseudo labels by including only high-confidence predictions (outputs with > 90%). To evaluate if errors during pseudo labeling generation would propagate to the model’s performance, we compare the performance of two approaches, with and without pseudo labels, in classifying cells. In this version (Figure 3A) we show that in data unseen during training, the addition of pseudo labels boosted the model performance; hence, resulting very unlikely a propagation of error during the pseudo label generation step that would have a significant impact of cell detection and classification.
> 5. Clarification of Scale in Figure 1:
> Thanks for feedback, unit is added as requested.

---

### Official Review · Reviewer_vYgz · 2024-03-04

**Confidence:** 4
**Preliminary Rating:** 1
**Final Rating:** 4

**Summary:**

In this paper, the authors introduce a method for detecting cell types in multiplex immunofluorescence (mIF) imaging. The approach first involves training a model, specifically YOLOv8, using a limited set of annotations. This trained model is then used to generate pseudo-labels, which are in turn used to retrain the model. The authors have conducted extensive experiments across various images representing different types of cancers and tissues.

**Strengths:**

- The authors have conducted experiments across a diverse range of cancer subtypes, accounting for tissue heterogeneity across different organs and cancer types.
- The paper addresses a notably challenging task of cell detection in multiple immunofluorescence (mIF) imaging, particularly within the constraints of limited annotations.

**Weaknesses:**

- The paper lacks a comparative analysis with state-of-the-art (SOTA) methods. For instance, the method could be compared with models such as Stardist, or Cellpose, pre-trained on immunofluorescence (iF) data and fine-tuned with the annotations provided here, with a simple thresholding method for cell classification. Furthermore, the performance of YOLOv8 trained solely on partial annotations is not discussed for all five cancer types in the evaluation dataset, which limits the ability to directly assess the interest of the proposed method compared to using the model trained on limited annotated data alone,  which appears to already exhibit satisfactory performance.
- The description of the experimental setup is missing critical details, such as the total number of pseudo-labels generated, the methodology used to tune this parameter, and the assessment of pseudo-label quality. The paper mentions strategies like Consistency-based Semi-supervised learning for object Detection (CSD) to prevent overfitting but fails to clarify if they were implemented or how they were adapted. The potential for overfitting is a significant concern with methods of this nature, and the results indicate some degree of overfitting, particularly in the "others" cell type across the four additional cancer types, raising important questions about the method's robustness.
- The clarity and structure of the paper could be improved, as it is currently challenging to follow. The blending of experimental settings with methods and results, coupled with a literature review that is not organized chronologically, hinders the reader's comprehension of the paper, and makes it difficult to identify clearly the contribution of the authors.

**Detailed Comments:**

N/A

**Justification Of Final Rating:**

The authors have responded to my concerns about the previous lack of comparisons by presenting baseline values derived from established cell segmentation methods, and about reproducibility by sharing their code. As pointed out by another reviewer, incorporating additional experiments that explore different semi-supervised strategies could further enhance the manuscript's quality.

**Justification Of The Preliminary Rating:**

At this current stage, the paper does not meet the required criteria for publication at MIDL2024, and the number of issues that urgently need to be addressed appears far too extensive for the allotted rebuttal period.

**Questions To Address In The Rebuttal:**

To qualify for publication at MIDL2024, the authors must improve the overall clarity and structure of the manuscript, clearly highlighting the novel contributions of their work. Additionally, it is imperative to undertake further experiments that compare their approach with state-of-the-art (SOTA) methods and evaluate the effectiveness of the model when trained solely on partial annotations on the evaluation datasets. These steps are essential to demonstrate the value and efficacy of the proposed semi-supervised learning method.

---

> ### Author Response · Authors · 2024-03-17
> **Clarifying of Our Approach: Comparative Analysis, Pseudo-Labels, Methodological, and Manuscript Structure revised**
>
> We thank Reviewer  for your constructive feedback. Our response aims to address the concerns raised, underlining the revisions made to enhance our manuscript's clarity, rigor, and comparative analysis depth.
>
> 1. Lack of Comparative Analysis with SOTA Methods:
> We thank the reviewer for the comment. Since the objective of this work was cell detection and classification, we focused on comparing against models developed for detection, such as Fast R-CNN, YOLOv5, and YOLOv8. Cellpos and Stardist are cell segmentation models and not directly applicable here. Stardist expects full cellular masks and it doesn't match our annotations. Without finetuning, the trained nuleus detection Stardist model applied to the blue channel in our mIF images (makers for nucleus), obtains an F1-Score of 0.35 in the validation dataset. We associate this poor performance with characteristics of mIF imaging such as poor definition of cell boundaries as well as small sizes of cells in our data. For finetuning starDist, we adjusted our labels by expanding the centroids of annotations by 50 pixels and creating circular masks for each annotation and achieved the average precision and recall scores of 0.284 across cell types. Given this poor performance compared to Yolov8 (Table 2, 0.848), we decided to not include these results in the manuscript as we believe this quick data preparation approach may have not been sufficient to provide StarDist a fair chance.
>
> 2. Performance of YOLOv8 with Partial Annotations:
> Thank you for your feedback. As suggested by the reviewer, to evaluate the contribution of pseudo labels to improve the task performance across the five cancer types, in this version we compare the performance of models with and without pseudo labels. We show that incorporating pseudo labels generation significantly improves F1 score (CD45: 0.656 vs. 0.949, panCK: 0.417 vs. 0.880, Others: 0.359 vs.0.740) with p values (Wilcoxon test) of 3.0517578125e-05, 1.9073486328125e-06, 3.0517578125e-05, respectively. We have incorporated these results in Figure 3A and section 3.2.
> 3. Experimental Setup and Overfitting Concerns:
> Thank you for this feedback. We have clarified the increase in dataset size after incorporating pseudo-labels: from initial counts of 11,643 for CD45, 15,228 for panCK, and 9,489 for Others, to 140,000, 200,000, and 120,000 for CD45, panCK, and Others, respectively. Also, we have updated Section 2.3 to address concerns about parameter tuning, pseudo-label quality, and overfitting. This procedure includes our method for creating pseudo-labels. We guaranteed the accuracy of these pseudo-labels by only accepting those with high confidence, that is, predictions with a confidence level of over 90%. This means new predicted labels were added only if their prediction confidence threshold was above 90%. As for mitigating overfitting we adopted strategies including dropout, data augmentation, regularization, and early stopping.  Furthermore, the model's performance was evaluated across five fully annotated cancer types not encountered during the training phase, resulting in average precision, recall, and F1 scores of 0.904, 0.851, and 0.848, respectively, for cell detection (calculated based on table 2 in manuscript). Also, the evaluation was conducted on a limited sample of four patches per cancer type. Considering the heterogeneity of tumors, this sample size may not have been sufficient to capture the full spectrum of variability present in each cancer type, potentially leading to an imbalanced testing set. This imbalance might have contributed to the lower performance observed in the 'Others' category for Urothelial Carcinoma and cholangiocarcinoma.
> 4. Clarity and Structure of the Paper:
> Acknowledging the feedback on the need for improved clarity and structure, we have undertaken a thorough revision of the manuscript. This includes a clearer separation of experimental settings, methods, and results, and a reorganized literature review to better highlight the novelty and contributions of our work. Also we removed some redundancy and These changes aim to facilitate a better understanding of our approach and its significance in the field of cell detection in mIF imaging.

---

> > ### Comment · Reviewer_vYgz · 2024-03-22
> >
> > Thank you for addressing my concerns and the additional experiments that compare the performance of the YOLOv8 model trained solely with partial annotations to your proposed method, as well as for detailing the steps taken to prevent overfitting. These additions indeed underline the merit of your approach. However, I maintain certain concerns regarding the comparative effectiveness of your strategy against state-of-the-art (SOTA) methods and the methodology behind constructing the pseudo-labels.
> >
> > The inclusion of experiments with Stardist is appreciated, yet it does not fully negate the potential relevance of segmentation methods for cell detection in this context. Despite Stardist's shortcomings in your tests, other methods mentioned in your introduction, like Cellpose or CellProfiler[1], applied on the blue channel, might offer better generalization to mIF imaging. Additionally, employing a basic approach with QuPath[2], followed by straightforward thresholding or leveraging a simple machine learning classifier trained on partial annotations for classification, could serve as a practical baseline.
> > Moreover, the Stardist results add value to the paper by demonstrating that the application of such methods is not always direct in your setting and underscores the rationale for focusing on detection-based methods like YOLO, which benefit from your novel training pipeline by requiring only partial annotations. However, further comparison with other baselines is necessary to fully justify the advantages of your proposed approach compared to currently used solutions.
> >
> > Regarding the generation of pseudo-labels, your explanations have clarified several points, yet some aspects remain confusing. In particular, you mention that the pseudo-labels were curated using a confidence threshold, however, Section 2.3 suggests a direct comparison of these pseudo-labels with ground truth annotations from pathologists ("We evaluated the generated pseudo labels through a designed loop by comparing them with the ground truth") which seems contradictory.
> >
> > Lastly, could you provide information on the availability of the code and data used in your experiments? This would greatly aid in the reproducibility of your results.
> >
> > [1]: Stirling DR, Swain-Bowden MJ, Lucas AM, Carpenter AE, Cimini BA, Goodman A (2021). CellProfiler 4: improvements in speed, utility and usability. BMC Bioinformatics, 22 (1), 433. . PMID: 34507520 PMCID: PMC8431850
> > [2]: https://qupath.readthedocs.io/en/stable/docs/tutorials/multiplex_analysis.html

---

> ### Author Response · Authors · 2024-03-27
> **The response highlights the inclusion of comparative analysis with SOTA methods and clarifies the generation and validation of pseudo-labels to address reviewers' concerns.**
>
> We thank Reviewer for your great feedback. Our response aims to address the concerns raised about SOTA and Pseudo label generating:
>
> 1. Lack of Comparative Analysis with SOTA Methods:
> Thanks for the great feedback and suggestions. We have improved our manuscript with additional experiments and discussions to address the concerns raised. We have included results from StarDist and Cellpose in Section 3.2 of our manuscript. These experiments were conducted applying their pre-trained models to our validation dataset. Specifically, we utilized only the blue channel for prediction and treated all cells as a single class to align with our research context. The performance metrics for these methods were evaluated by aligning the centers of predicted segmentation masks with the bounding boxes of annotated cell centers, acknowledging the absence of segmentation masks in our ground truth.
> The results from StarDist and Cellpose are now detailed in the manuscript, indicating a precision of 0.450, a recall of 0.210, and an F1 score of 0.284 for StarDist, whereas Cellpose achieved a precision of 0.862, a recall of 0.230, and an F1 score of 0.363. These findings confirm our decision to focus on detection methods, like YOLO, optimized through our novel training pipeline with only partial annotations. Furthermore, they underscore the challenges and conserns when applying segmentation methods to our specific dataset.
> In the discussion and limitations sections, we have expanded upon the implications of these comparisons. We acknowledge our primary aim of identifying cell centers and detection and juxtaposing our approach with state-of-the-art methods,  StarDist and Cellpose, validateing our methodology's efficacy and opening routes for future research. This includes integrating segmentation masks to enhance cell annotations and improving comparative performance against leading segmentation methods.
> This addition provides a comprehensive justification for our approach, emphasizing its advantages over existing solutions while acknowledging the constructive potential for methodological enhancements in future studies.
>
> | Method   | P     | R    | F1    |
> |----------|-------|------|-------|
> | Stardist | 0.450 | 0.210| 0.284 |
> | Cellpose | 0.862 | 0.230| 0.363 |
> | YOLOv8s  | 0.904 | 0.850| 0.847 |
>
> 2. generation of pseudo-labels: hank you for your feedback. To clarify and address the concerns of potential contradictions as you highlighted, we have revised our explanation as follows: Upon generating new labels, we initially applied a confidence threshold of 90%. Among these labels, we ensured the inclusion of all labels that align with the ground truth or partial annotations provided by pathologists, in addition to the new labels generated. This verification process assures us of the reliability of our pseudo labels, ensuring we incorporate only the most dependable ones into our dataset. Here is full modified sentences in our manuscript:
>
>  “In our approach, pseudo labels were first generated by selecting predictions with a confidence level above 90\%. These high-confidence pseudo labels were ensured to contain almost all initial partial annotations from pathologists. This process maintained pseudo labels with high accuracy, acting as an additional verification step. Collectively, the method used a confidence threshold for the initial generation and ground truth comparison for validation, ensuring that only the most reliable pseudo labels were added to the final training dataset”
>
> 3. data availability:
> Thanks for asking about our code and data. The data from our study cannot be fully shared right now and can be deposited into the public repository only after the publication of the clinical study because of some internal policies we need to follow. But we have still managed to do a couple of things:
> - We have included some samples of our dataset in GitHub. Highlighting our patches and annotations formats.
> - We have put all the related code on GitHub for anyone to look at and use.
>
>
> Thanks for understanding. We are all for sharing our science as openly as possible within these guidelines.
> Our GitHub repository: https://github.com/idso-fa1-pathology/semi-supervised-cell-detection

---

### Meta-Review · Area_Chair_i1R2 · 2024-03-30

**Recommendation:** Accept (Poster)
**Confidence:** 4

**Metareview:**

The majority of the reviewers render positive ratings.

The evaluation of the work revolves around an innovative method for detecting cell types in MxIF imaging, leveraging a semi-supervised learning approach with YOLOv8. The authors addressed a challenge in cell detection within mIF imaging, focusing on improving model training with limited annotations through the generation of pseudo-labels. This method was tested across various cancer types, showcasing its potential to enhance the accuracy of cell detection in complex biological samples.

[pros]
The semi-supervised learning model using YOLOv8 and the generation of pseudo-labels for training with limited annotations is novel in the context of MxIF imaging for detecting different cell types across various cancer tissues. Conducting experiments across a diverse range of cancer subtypes demonstrates the method's robustness and adaptability to tissue heterogeneity. The authors' approach improved the F1 scores across different cell types, demonstrating the efficacy of incorporating pseudo-labels into the training process.

[cons]
The paper falls short in providing a comprehensive comparison with SOTA methods, limiting the ability to fully assess the novelty and efficacy of the proposed method against existing solutions. The paper lacks crucial details about the experimental setup, including the total number of pseudo-labels generated and the methodology for tuning this parameter. The concern of overfitting, especially with semi-supervised methods that rely heavily on pseudo-labels, was not adequately addressed.

---

### Decision · Program_Chairs · 2024-04-05

Accept (Poster)